# Morquio A Syndrome: Identification of Differential Patterns of Molecular Pathway Interactions in Bone Lesions

**DOI:** 10.3390/ijms25063232

**Published:** 2024-03-12

**Authors:** J. Victor. Álvarez, Susana B. Bravo, María Pilar Chantada-Vázquez, Carmen Pena, Cristóbal Colón, Shunji Tomatsu, Francisco J. Otero-Espinar, María L. Couce

**Affiliations:** 1Department of Forensic Sciences, Pathology, Gynecology and Obstetrics, Pediatrics, Neonatology Service, Health Research Institute of Santiago de Compostela (IDIS), Hospital Clínico Universitario de Santiago de Compostela, CIBERER, MetabERN, 15706 Santiago de Compostela, Spain or jose.victor.alvarez.gonzalez@sergas.es (J.V.Á.); cristobal.colon.mejeras@sergas.es (C.C.); 2Health Research Institute of Santiago de Compostela (IDIS), CIBERER, MetabERN, 15706 Santiago de Compostela, Spain; 3Proteomic Platform, Health Research Institute of Santiago de Compostela (IDIS), Hospital Clínico Universitario de Santiago de Compostela, 15706 Santiago de Compostela, Spain; sbbravo@gmail.com (S.B.B.); mariadelpilarchantadavazquez@gmail.com (M.P.C.-V.); carmen.pena.pena@sergas.es (C.P.); 4Skeletal Dysplasia Lab Nemours Biomedical Research, Nemours Children’s Health, 1600 Rockland Road, Wilmington, DE 19803, USA; stomatsu@nemours.org; 5Paraquasil Platform, Health Research Institute of Santiago de Compostela (IDIS), Hospital Clínico Universitario de Santiago de Compostela, 15706 Santiago de Compostela, Spain; francisco.otero@usc.es; 6Department of Pharmacology, Pharmacy and Pharmaceutical Technology, School of Pharmacy, Campus Vida, University of Santiago de Compostela, 15872 Santiago de Compostela, Spain

**Keywords:** animal studies, biomarkers, mucopolysaccharidosis type IV, musculoskeletal manifestations, proteomic

## Abstract

Mucopolysaccharidosis type IVA (MPS IVA; Morquio A syndrome) is a rare autosomal recessive lysosomal storage disease (LSD) caused by deficiency of a hydrolase enzyme, N-acetylgalactosamine-6-sulfate sulfatase, and characterized clinically by mainly musculoskeletal manifestations. The mechanisms underlying bone involvement in humans are typically explored using invasive techniques such as bone biopsy, which complicates analysis in humans. We compared bone proteomes using DDA and SWATH-MS in wild-type and MPS IVA knockout mice (UNT) to obtain mechanistic information about the disease. Our findings reveal over 1000 dysregulated proteins in knockout mice, including those implicated in oxidative phosphorylation, oxidative stress (reactive oxygen species), DNA damage, and iron transport, and suggest that lactate dehydrogenase may constitute a useful prognostic and follow-up biomarker. Identifying biomarkers that reflect MPS IVA clinical course, severity, and progression have important implications for disease management.

## 1. Introduction

Mucopolysaccharidosis type IVA (MPS IVA; Morquio A syndrome) (OMIM 253000) is a rare autosomal recessive lysosomal storage disease (LSD) caused by deficiency of a hydrolase enzyme, N-acetylgalactosamine-6-sulfate sulfatase (GALNS, EC 3.1.6.4) [1,2], which results in the accumulation of glycosaminoglycans (GAGs) such as keratan sulfate (KS) and chondroitin-6-sulfate (C6S) in multiple tissues, primarily bone, cartilage, heart valves, and cornea, leading to devastating systemic skeletal dysplasia with incomplete ossification and consequent growth impairment [3,4].

The prevalence of MPS IVA ranges from 1 per 76,000 births in Northern Ireland to 1 per 640,000 births in Western Australia [5]. Excessive accumulation of KS and C6S in bone, cartilage, and their extracellular matrix (ECM) causes a unique skeletal dysplasia in patients with MPS IVA. Although most MPS IVA patients generally appear healthy at birth, skeletal deformities tend to manifest in the first few years of life. Common features in severe MPS IVA include skeletal dysplasia with short neck and trunk, cervical spinal cord compression, tracheal obstruction, pectus carinatum, joint laxity, kyphoscoliosis, coxa valga, and genu valgum [6,7]. MPS IVA patients often become severely disabled and require a wheelchair in adolescence. Marked non-skeletal manifestations, including respiratory disease, spinal cord compression, cardiac disease, impaired vision, hearing loss, and dental problems, are also described in MPS IVA patients [8,9]. Untreated patients with severe disease typically die in the second or third decade of life due to respiratory problems, cervical spinal cord complications, or heart valve disease [7,10].

While musculoskeletal manifestations predominate in MPS IVA, there are certain similarities in the underlying processes with more common bone diseases. Alterations observed at the cartilage level include abnormal structure and/or degradation of the intra- and extracellular matrix, organelle malfunction, vacuolated cells, and cell death, leading to reduced numbers of chondrocytes, erosion of articular cartilage, and bone growth impairment. Although the clinical presentation of MPS IVA is well documented, the mechanisms underlying the associated skeletal abnormalities remain to be fully elucidated. The changes that occur are associated with an inflammatory response and altered gene and protein expression. This response is similar to that seen in arthritis (AR) and osteoarthritis [11,12].

While some aspects of the pathology are described [13,14], analyses are complicated by difficulties in acquiring the necessary samples, which require invasive bone biopsy techniques. In recent years, proteomics approaches using human bone cells have been used to identify biomarkers and/or bone lesion patterns in other lysosomal diseases [15,16]. However, until now, no such studies have analyzed bones from MPS IVA patients or animal models. Using two proteomic technologies, we investigated differential patterns of molecular pathway interactions by analyzing femur bones from age- and sex-matched MPS IVA and wild-type (WT) mice. The mice examined were aged 6–8 weeks, which represents an intermediate stage of disease progression in which KS deposits are already detectable in the bone. Only male mice were selected, as the hormonal changes that occur in female mice represent potential confounding factors.

## 2. Results

Femur bones from untreated UNT mice [17] and from WT mice were extracted and analyzed using two precise and sensitive proteomic techniques: DDA (data-dependent analysis) and SWATH (sequential window acquisition of all theoretical fragment ion spectra), an information-dependent acquisition (IDA) technique. 

DDA identified 1056 and 1227 proteins in UNT and WT mice, of which 129 were exclusive to the UNT group and 300 to the WT group (Appendix A). SWATH-MS identified 900 proteins and quantified 618 proteins, ultimately identifying 184 differentially expressed proteins (DEPs) (Appendix A). 

We initially focused on proteins identified by DDA for which SWATH-MS revealed differential expression between UNT and WT mice.

### 2.1. Section I: Dysregulation of Metabolic Pathways Implicated in Energy Production

Cells produce ATP using three main interconnected metabolic pathways: glycolysis, the *Krebs* cycle (TCA cycle), and oxidative phosphorylation (OXPHOS). 

We identified around 100 proteins related to energy production (Figure 1A). STRING analysis revealed three large clusters of proteins related to glycolysis, gluconeogenesis, TCA cycle, amino acid metabolism, fatty acid β-oxidation, and the electron transport chain (Figure 1B). Together, SWATH-MS and DDA identified 77 proteins. In comparison, DDA alone identified 22 (Figure 1C). 

DDA identified 15 proteins involved in glycolysis (Appendix A). SWATH-MS showed that eight were significantly upregulated in the UNT group (*p* < 0.05). Aldoa, Eno 3, and Ldha showed a 3-fold increase relative to the WT group (Figure 2A; Appendix A). Only Eno 1 was downregulated more than 3-fold in UNT mice. Of 14 proteins involved in the TCA cycle (Appendix A) that DDA identified, Pdhb was downregulated in UNT versus MT mice, while Idh3a and Mdh2 were upregulated with a fold change (FC) > 3 (Figure 2B; Appendix A).

Mitochondria are key players in cellular energy regulation and can produce most of the required adenosine triphosphate molecules via the oxidative phosphorylation system (OXPHOS). DDA identified several proteins involved in OXPHOS (Appendix A): 30 corresponded to complex I, including Ndufa6, Ndufa13, Ndufb9, Ndufc2, and Ndufv, all of which were upregulated with an FC > 3 in UNT versus WT mice (Figure 2C). Ndufb4 was detected in the UNT group only. Three proteins corresponding to complex II, including Sdha and Sdhb, were significantly upregulated in UNT, with Sdhb having an FC > 3 (Figure 3A). Of the five complex III proteins identified by DDA analysis, only one (Uqcrfs1) was significantly upregulated in UNT versus WT mice. DDA identified nine proteins corresponding to complex IV (Appendix A). Of these, four were significantly upregulated in UNT mice, with an FC > 3 for Cox5a, Cox6a2, and Cox7a2 (Figure 3B). Mitochondrial (mt) ATP synthase or complex V uses energy from the electrochemical proton gradient to phosphorylate ADP, producing ATP. DDA identified 14 proteins (Appendix A) related to this process: 6 were significantly upregulated in the UNT group (Atp5f1d, Atp5mf, and Atp5po with an FC > 3), while Atp5mk was downregulated (Figure 3C; Appendix A).

Of proteins related to β-oxidation that were identified by DDA (Appendix A), Acadvl and Fabp3 were significantly upregulated in UNT with an FC > 3.0, while Eci1 was downregulated (Figure 3D; Appendix A).

In summary, the upregulation of several proteins involved in energy production in UNT mice suggests high energy production in this group. 

### 2.2. Section II: Proteins Related to Mitochondria Function: Oxidative Stress, Voltage-Dependent Channels, Apoptosis, Compensatory Mechanisms of Oxidative Stress, and DNA Damage

DDA identified 43 proteins related to oxidative stress, voltage channels, apoptosis, and DNA damage and repair (Figure 4A and Appendix A). STRING analysis revealed proteins related to apoptosis, transport, iron transport, and DNA damage (Figure 4B). SWATH-MS and DDA identified 32 proteins, while DDA alone identified 11 (Figure 4C).

DDA identified 15 proteins related to reactive oxygen species (ROS) (Appendix A). On SWATH-MS, eight were significantly upregulated in UNT versus WT mice (FC > 3 for Sod1, Prdx1, Prdx2, Prdx3, Prdx5, and S100a9; Figure 5A; Appendix A). Six proteins related to voltage-dependent channels in mitochondria were identified by DDA, none of which were significantly dysregulated in UNT mice, although all showed minor expression in UNT mice. DDA detected five proteins related to apoptosis in both groups, four of which showed no significant differences between UNT and WT mice. Casp6 was identified in the WT group, suggesting a defect in apoptosis in UNT mice (Appendix A). Of 15 detected proteins related to iron transport (Appendix A), 6 (Tf, Cisd1, Fth1, Mb, Hbb-b1, and Hpx) were significantly upregulated in UNT versus WT mice, with FC > 3 (Figure 5B and Appendix A). DDA identified tree proteins (TPi1, Uba, Ube2n) related to the prevention of DNA damage (Appendix A), all of which were upregulated in UNT versus WT mice with FC > 3 (Figure 5C; Appendix A).

Our findings indicate a marked elevation of oxidative stress in UNT mice, mainly via Prdx3/Prdx5. UNT mice showed reduced expression of proteins related to voltage-gated channels and no activation of the apoptotic pathway via caspases. However, UNT mice also showed marked upregulation of proteins involved in iron capture and increased expression of proteins that participate in DNA damage prevention.

### 2.3. Section III. Histones, Ribosomes, Proteasomes, Vesicular Transport, and Lysosomes

We identified 186 proteins related to histones, ribosomes, proteasomes, vesicular transport, and lysosomes (Appendix A). The Reactome analysis (Figure 6A) depicts proteins related to lysosomes (part I) and vesicle transport (part II). A String analysis revealed clusters of proteins related to ribosomal organization and biogenesis, proteasomes, lysosomes, and signal translation (Figure 6B). Together, SWATH-MS and DDA identified 83 proteins, while DDA alone identified 103 proteins (Figure 6C).

DDA identified two integrins that regulate the nucleus-to-cytoplasm trafficking of macromolecules (Kpna2, Kpnb1) in WT mice only (Appendix A). On SWATH-MS, five histones were upregulated in UNT versus WT mice with an FC > 3 (H1-0, H1-2, H1-4, H2az2, and H4c1) (Figure 7A; Appendix A). 

Ribosomal proteins: DDA identified 22 proteins from the 40S ribosomal subunit (Appendix A), of which 5 were significantly upregulated in the UNT group, 3 with an FC > 3 (Rps5, Rps18, and Rps25) (Figure 6B; Appendix A). Of the 27 detected proteins related to the 60S region, only 1 (Rpl12) showed an FC > 3 in UNT versus WT mice (Figure 7B; Appendix A). Seven ribosomal proteins were detected only in UNT mice (Rpl6, Rpl18, Rpl18a, Rpl24, Rpl27, Rpl28, and Rpl38) (Appendix A).

Proteasomes: DDA identified 22 proteasome proteins (Appendix A), only 1 of which was significantly upregulated in the UNT group with an FC < 3 (Appendix A). Psmd8 was expressed only in UNT mice, while 12 proteins were expressed only in WT mice (Psma2, Psma3, Psma6, Psma7, Psmb1, Psmb2, Psmb3, Psmb5, Psmb6, Psmb8, Psmb9, and Psmd2) (Appendix A). 

Vesicular transport: DDA identified 30 proteins related to vesicular transport (Appendix A), 3 of which were significantly upregulated in UNT mice, 2 with an FC > 3 (Rab5b, Tpd52l2) (Figure 7C; Appendix A). Sorl1 was detected only in UNT mice. Of 13 detected proteins related to the lysosomal membrane (Appendix A), 4 were significantly upregulated in UNT mice, 3 with an FC > 3 (M6pr, Lgals1, and Lgals3) (Figure 7D; Appendix A). Cd63, Cd74, and Atp6v1a were expressed only in WT mice. Of the 30 proteins identified, 22 were in the lysosome (Appendix A), 1 of which was downregulated in UNT mice with an FC < 3 (Asah1) (Figure 7E; Appendix A). It should be noted that Galns (of which a deficit is observed in MPS IVA) and other related proteins (Gba1, Pla2g15, and Naaa) were identified only in WT mice.

DDA identified 44 cytoskeleton-related proteins (Appendix A), 17 of which were identified as DEPs on SWATH-MS: 5 were upregulated in WT versus UNT mice, 4 with an FC < 0.3 (Calm3, Acta1, Tnnc2, and Gdpd2) (Figure 8A; Appendix A). Eleven proteins were upregulated in UNT versus WT mice, seven with FC > 3 (Myl2, Myfpl, Lasp1, Rhoa, Arpc3, Arpc5, and Arpc5l) (Figure 8B; Appendix A).

In summary, in UNT mice, we observed an increase in protein biosynthesis via histones and ribosomes, a partial lack of proteasome proteins, a lack of calcium necessary for cytoskeletal proteins, an increase in proteins related to vesicular trafficking, the presence of protein-related lysosomal membrane damage repair, and an absence of certain membrane receptors. Crucially, we observed a total absence of the Galns enzyme in UNT mice, which may result in the accumulation of KS and C6S in lysosomes (KS and C6S are sugar chains, which were not measured in the present study), a characteristic feature of MPS IVA disease.

### 2.4. Section IV. Extracellular Matrix

We identified 99 proteins related to cell adhesion (Figure 9A, Appendix A). SWATH-MS and DDA identified 60 proteins, while DDA alone identified 39 (Figure 9B).

To facilitate interpretation of the large numbers of matrix components identified, proteins were ordered by families, as indicated below. 

Hyaluronan-related proteins: a single protein (Cemip2) was identified (Appendix A) but showed no significant difference in expression between WT and UNT mice. 

#### Proteoglycans

Hyalectans: three proteins were identified. Hapln1 was significantly upregulated in UNT versus WT mice (FC < 3). Small leucine-rich proteoglycans: DDA identified the following *Class I*, *II*, and *IV proteins* (Appendix A):
○*Class I proteins*: Dcn was significantly upregulated in UNT mice (FC > 3; Figure 9C, Appendix A). Aspn was detected in UNT mice only. ○*Class II proteins*: DDA identified four proteins (Fmod, Lum, Prelp, Kera; Appendix A), all of which were significantly upregulated in UNT mice with FC > 3 (Figure 9C, Appendix A). ○*Class IV*: Only Chad was significantly upregulated in UNT mice, with an FC < 3 (Appendix A). Cell surface proteoglycans: DDA identified proteins related to the cell surface (e.g., Cspg4; Appendix A), none of which were differentially expressed in UNT versus WT.

DDA identified various collagen proteins, which we classified according to physiological function (Appendix A) as follows:**Fibrillar collagen**: DDA identified eight proteins related to fibrillar collagen, five of which were significantly upregulated in UNT mice, four with an FC < 3 (Col1a1, Col1a2, Col11a1, and Col11a2) (Figure 9D, Appendix A). Col5a1 was identified only in WT mice.**Networking-forming collagen**: three proteins were identified, one of which was present only in WT mice (Col4a1).**FACITs, beaded-filament-forming collagens:** Three proteins identified.**MACITs, anchoring fibrils, multiplexin:** One protein identified.

**Laminins:** Five proteins were identified, none of which differed significantly between groups.

**Matrix cellular proteins:** DDA identified 13 proteins related to the ECM, 2 of which were significantly upregulated in UNT mice, 1 with an FC > 3 (Thbs1) (Figure 10A, Appendix A). Cnn3 was detected only in WT mice (Appendix A). 

**Cellular interface:** DDA identified two proteins, which were present only in WT mice.

**Extracellular matrix proteases:** DDA identified three proteins that can be classified in the **Metzincin** family. Mmp9 was significantly upregulated in UNT mice, with an FC > 3 (Figure 10B, Appendix A). Adam10 was detected only in UNT mice. 

**Plasminogen/plasmin system:** DDA identified 15 proteins (Appendix A), 3 of which were significantly upregulated in UNT mice, 1 with an FC > 3 (Serpinf1) (Figure 10C; Appendix A). Plaur and Serpina1 were identified only in WT mice. 

**Cathepsin proteases:** DDA identified two proteins in the *Aspartic proteases* family (Appendix A), one of which was upregulated in UNT mice, with an FC > 3 (Ctsd) (Figure 10D; Appendix A). 

***Serine proteases:*** One protein identified (Ctsg). 

***Cysteine proteases:*** DDA identified three proteins, one of which (Ctsh) was upregulated in UNT mice with an FC > 3 (Figure 10D; Appendix A). 

**Extracellular receptor matrix**: DDA identified 14 proteins (Appendix A) belonging to the Integrin family, 4 of which were significantly up- or downregulated in UNT mice, but only 2 with an FC > 3 (Itgam, Itgav) (Figure 10E; Appendix A). 

**Discoidin domain receptors:** DDA identified two proteins, of which only Cdc42 was upregulated in UNT mice (FC < 3; Appendix A). DDR2 was identified only in WT mice.

**Another extracellular receptor:** DDA identified three proteins, one of which (Dpep1) was upregulated with an FC > 3 (Figure 10F; Appendix A). 

In summary, our analysis revealed significant upregulation in UNT mice of several proteoglycans (N-linked glycosylation), mainly related to KS (Fmod, Lum), metalloproteases, cathepsins, and collagens (some fibrillar collagens). 

### 2.5. Section V. Other Proteins of Interest

We identified five other proteins of interest (Rtn2, Fetub, Alb, Ckm, and Ckmt2; Appendix A), four of which were significantly upregulated in UNT mice with an FC > 3 (Figure 11; Appendix A).

### 2.6. Quantitative Validation of DEPs

#### 2.6.1. Mouse DEPs Validation by Different Proteomic Technology

To perform cross-validation of DEPs identified by quantitative SWATH-MS (MS2-based quantification), we used Scaffold software v 5.3.2 to obtain a semiquantitative value for proteins identified by label-free assay based on spectral count. This label-free method enabled MS1-based quantification (spectral count). We observed a correlation between the two quantitative methods of approximately 18.90% (Appendix A), suggesting the quantitative reliability of the proteins considered DEPs in our study. The number of DEPs identified by SWATH-MS was identical to that identified by spectral count (121 DEPs). Using the two quantitative methods, we validated 38 proteins (Figure 12A; Appendix A). Among these proteins, we identified lactate dehydrogenase (LDH) Ldha and Ldhb as potential biomarkers (Figure 12B; Appendix A). STRING analysis of the validated DEPs identified proteins related to the ATP metabolic process, glycolysis and gluconeogenesis, TCA cycle, LDH, OXPHOS, and ROS as potential biomarkers.

#### 2.6.2. In Silico Validation of Mouse DEPs Using Data Sets from Human Samples

To validate our quantitative proteomic data obtained by SWATH-MS in mice, we compare the quantitative proteomic data obtained by SWATH-MS in mice with human data previously obtained by our group using the same proteomic technology [18,19]. As shown in Figure 13A and Appendix A, several detected proteins were common to both analyses, including proteins involved in pathways such as glycolysis, oxidative stress, and vesicle trafficking and proteins involved in the cellular cytoskeleton. Perhaps most interestingly, Ldha and Ldhb were common to all analyses, supporting the potential of these proteins as candidates for diagnosis, prognosis, and disease monitoring biomarkers.

It should be noted that although Morquio A is a rare disease with few studies, our findings point to multiple similarities between Morquio A patients and those with other common osseous diseases of adults, including arthritis. Therefore, the proteomic approach used here to identify potential therapeutic targets for this rare disease could also help further our understanding of other bone diseases. For this reason, proteomic data obtained from human osteoarthritis bone samples were selected for in silico validation of our findings.

The presence of multiple proteins common to our mouse data set and the two human data sets validates the approach used in the present study. The first data set consisted of bone marrow sample data taken from Hennrich et al. [20] and corresponded to 59 human subjects (45 male, 14 female). In that study, relative quantification was achieved using tandem mass tag (TMT) labeling, SWATH-MS. The second data set consisted of data from primary cultures of human articular chondrocytes, from both healthy and osteoarthritic patients, taken from a study by Konstantinos et al. [21]; in that paper, samples were analyzed using label-free quantification and scaffold software v 5.3.2 used for comparative analysis, as in the present study. Comparison of both data sets with our data confirmed that the protein alterations found in our mouse model resembled many of those observed in a bone disease such as osteoarthritis. 

Comparison of our findings with those of the 2 aforementioned human studies revealed 30 proteins common to all 3 analyses (Figure 13B and Appendix A) as well as comparable alterations in pathways implicated in glycolysis, ROS, DNA damage, vesicle transport, lysosomes, the cellular cytoskeleton, and the extracellular matrix (moreover, Ldha and Ldhb were among the common proteins across all 3 studies). Other candidate biomarkers detected in all three data sets included collagen type II and MMP9.

In summary, our “in silico” validation results indicate that the DEPs found in mice in the present study are also observed in human samples, particularly MMP9, Col2, and both Ldha and Ldhb. 

## 3. Discussion

The overall goal of the present study was to investigate the differential expression of proteins involved in distinct molecular pathways in MPS IVA and to explore the pathogenic relevance of the observed differences in systemic skeletal dysplasia in this disease. To this end, we investigated the differential expression of proteins of interest in the bone ECM of MPS IVA mice. Due to the rarity of this disease, it is challenging to obtain sufficient numbers of bone samples from affected human patients and published studies using proteomics of human bone samples are very scarce. Therefore, we based our analysis on samples from an MPS IVA mouse model and compared the results obtained with SWATH-MS proteomic data previously published by our group corresponding to fibroblasts and blood cells from MPS IVA patients with published data from proteomic analyses of bone marrow and bone (primary chondrocytes) [20,21] samples from humans. Moreover, we linked the altered proteins and pathways observed in MPS IVA mice with the multiple causes affecting MPS IVA patients. In our analysis, we sought to identify candidate diagnostic and prognostic biomarkers for MPS IVA that were validated in humans in our previous articles. 

MS-based proteomics is an indispensable tool for biomarker discovery. We used two distinct but complementary proteomic technologies, SWATH and DDA analysis, to better understand MPS IVA. The advent of SWATH-MS has ushered in a new era of accurate, reproducible label-free proteome quantification [22,23]. Our group previously used this technology to characterize cells and plasma/serum samples from mice models of MPS [18,19,24]. Unlike conventional DDA-MS, which involves the specific selection and fragmentation of a limited number of precursor ions (typically the most abundant ones) in a survey scan, SWATH-MS is based on the cyclical acquisition of precursor ions within specific isolation windows covering the entire *m*/*z* range. Spectral counting is widely used for proteome quantification because it is a simple, label-free method [25].

Using DDA and SWATH-MS, we characterized differentially expressed proteins in bone ECM in MPS IVA mice. Our findings demonstrate alterations in specific cellular functions in the ECM of bone samples from UNT mice. The proteins analyzed include those related to metabolic pathways (i.e., cytoskeleton proteins and proteins involved in mitochondrial function, autophagy, interaction of calcium, iron, and other ions, ribosomes, lysosomes, and the ECM). We observed altered expression of proteins involved in energy-generating pathways, including glycolysis, the Krebs cycle, and OXPHOS, which produce high oxidative stress. Furthermore, we detected altered expression of proteins such as Rtn2 and Sorl1 that facilitate glucose entry into cells via GLUT-4 [26,27]. Similar alterations in osteoarthritis in humans are described, in which GLUT-1 receptor expression is augmented, thereby increasing cellular glucose levels [28]. These high glucose levels result in significant alterations in the expression of other proteins in UNT mice, in which higher levels of cellular hypoxia are observed [29]. Alterations in metabolic pathways also increase lactate production, as evidenced by the upregulation of LDH observed in UNT mice. LDH plays a vital role in generating oxidative stress in rheumatoid arthritis. Using mRNA sequencing in primary chondrocyte cultures from a mouse model of osteoarthritis, Arra et al. [30] detected significant increases in the expression of genes involved in glycolysis, including Ldha. We recently identified candidate biomarkers of MPS IVA [18,24] using the same proteomic technologies in two different sample types: fibroblasts and leukocytes from MPS IVA patients. Those studies showed that LDH expression is upregulated in MPS IVA cells and that levels decrease in response to enzyme replacement treatment [30,31], providing strong evidence to suggest a role for LDH proteins as biomarkers of disease prognosis and progression.

Another important finding is the general upregulation of proteins related to the Krebs cycle and OXPHOS in UNT mice. We detected many upregulated proteins pertaining to mitochondrial complexes I, II, III, IV, and V (ATP-synthetase), including proteins such as Ndufa6 with an FC 86.51, corroborating previous reports of increased ROS levels [30]. Furthermore, we observed elevated expression of ROS compensatory proteins such as Prdx3/Prd [32]. In line with previous findings in an animal model of MPS IVA, upregulation of DNA damage repair proteins has been reported in UNT mice [33]. In that paper, Donida et al. analyzed blood samples from 17 MPS IVA patients and 14 healthy controls using molecular biology techniques and observed increased DNA damage of oxidative origin in patients versus controls. Moreover, they detected increased interleukin 6 (IL-6) expression in patients, suggesting a possible link between inflammation and oxidative stress in MPS IVA. Although we detected proteins related to inflammation in our study, the proteomic techniques used did not allow the detection of low-molecular-weight proteins such as IL-6, despite its likely presence in our samples. These alterations in inflammation pathways result in activation of signaling pathways such as STAT3 [34], a transcription factor activated by members of the IL-6 and IL-10 family and by IL-21, IL-27, G-CSF, leptin, and IFN-I, among other signaling molecules [35]. In their mass spectrometry studies of an AEP^−/−^ (Asparaginyl endopeptidase knockout) mouse model and cell lines, Martínez-Fábregas et al. [36] concluded that an increase in lysosomal cysteine proteases (CtsD) in the kidney reflects the accumulation of undegraded material and/or increased expression of lysosomal proteins. Moreover, they reported an increase in endocytic substrate load, independent of the transcription factor EB (TFEB) and triggered by STAT3 activation downstream of lysosomal oxidative stress. Similar lysosomal adaptations are seen in mice and cells expressing a constitutively active form of STAT3 [36]. Therefore, Martínez-Fabregas J et al. reported that STAT3 signaling is activated by the ROS effect due to substrate accumulation in lysosomes and deficiencies in lysosomal proteases [36] which also occurs in MPS IVA.

The protein MTORC1-TFEB acts as a transcriptional regulator of autophagy and is the most relevant marker of lysosomal activity [37,38] owing to its ability to activate lysosomal biosynthesis. Our data indicate that this protein appears to be inhibited by overexpression of cathepsin D and LAMP1 in MPS IVA. These proteins also contribute to increased inhibition of the autophagy pathway via the lysosome. Proteosomes play a key role [39,40] in this autophagy pathway, mainly via Psma6 and Psmb6, which participate in the active nucleus of the proteasome [41]. However, we did not detect proteins related to proteasomes in UNT mice. Activation of STAT3 [39] in response to lysosomal accumulation stimulates compensatory biosynthesis of cytokines and interleukins. The cytokines and interleukins detected in the present study were the same as those described in other MPS IVA studies. The authors of [42] used liquid chromatography–tandem mass spectrometry (LC–MSMS) to measure the 4 disaccharides produced from GAGs in serum or plasma from 34 MPS IVA patients and identified 9 biomarkers that were significantly elevated in untreated MPS IVA patients: EGF, IL-1β, IL-6, TNF-α, MIP-1α, MMP-1, MMP-2, and MMP-9. In UNT mice, we observed activation of protein biogenesis, which explains the high levels of cytoplasmic histones (some of which have been previously detected in proteomic studies of other non-bone pathological cells [18,19]) and upregulation of ribosomal proteins and the exclusive presence of specific proteins such as Rps19 and Rpl24, in agreement with previous reports [43]. This increase in protein biosynthesis normally generates more proteins with endosomal/lysosomal functions [44] and increases ribosome expression in the bone, which usually occurs in the context of low TFEB activity [45].

Analysis of proteins involved in vesicular transport revealed alterations in the endocytosis pathway in UNT mice, as evidenced by upregulation of Rab5c, which participates in vesicular trafficking [38], and Tpd52l2 [46]. This pathway is important because increases in M6pr can induce the expression of many lysosomal enzymes [47]. In the UNT group, we detected overexpression of proteins related to lysosomal membrane repair and autophagy functions [48]. Conversely, other lysosomal membrane proteins, such as Atp6v1a, Cd63, and Cd74, were absent from UNT mice.

In analyzing apoptosis activation, we found that caspases that would normally be elevated in response to high ROS levels [49] were not expressed at high levels in UNT mice. Moreover, the calcium voltage potential between the internal and external membrane was reduced (in UNT mice), which can prevent activation of apoptosis functions [50,51], suggesting a problem affecting calcium pathways. In the cytoplasm, we observed clear evidence of dysregulation of actin (Acta 1), myosin (Myosin-1), calmodulin (Calm3), and troponin (tnn3) [52,53] in UNT mice. These alterations may have consequences for cytoskeletal protein function. Moreover, ROS may also contribute to the failure of repair mechanisms by reducing the capacity of chondrogenic precursor cells to migrate and proliferate within an injured area: NO has been shown to inhibit chondrocyte migration and attachment to fibronectin via modification of the actin cytoskeleton [54]. This is particularly important in the context of MPS IVA because bone resorption proteins that form the osteoclast podosome require proteins such as F-actin-rich domain, protein 2/3 complex (Arp2/3), and myosin light chain to attach to the bone area for bone degradation (in the UNT group, we found several proteins elevated) (Arpc3, Arpc5, Arpc5l, Capza2, Myfpl, Myl2) [55,56]. Actin-binding points must be in good condition to enable the transport of lysosomes to the matrix. This does not occur in UNT mice, in which filamin B (Flnb) expression is altered, as described previously by Qiming Xu et al. [57]. Those authors reported that decreased expression of this protein could induce skeletal malformations, including delayed ossification in long bone growth plates; reduced bone mineral density; dysregulation of muscle differentiation; intervertebral disc ossification; altered chondrocyte proliferation, differentiation, and apoptosis; impairment of angiogenesis; and hypomotility of osteoblasts, chondrocytes, and fibroblasts [57]. Moreover, lysosomes can transport calcium phosphate [58], obtained by collecting calcium from other organelles such as the endoplasmic reticulum, Golgi, cytoplasm, and mitochondria. Normally, this transport occurs via kinesin motor and dynein pathways [59]. In the present study, the proteins related to the dynein pathway were detected only in WT mice. The transport of collagen and calcium phosphate, which was also altered in UNT mice, is vital in supporting the role of osteoblasts in bone mineralization.

In our study in UNT mice, the evident decompensation ROS production induces degradation of the bone ECM. Bake et al. [60] reported that exposure of chondrocytes to H_2_O_2_ inhibits proteoglycan. Findings reported by Taskiran et al. [61] in rabbit articular cartilage and by Häuselmann et al. [62] (in human articular chondrocytes) suggest that interleukin-1 suppresses the synthesis of the cartilaginous matrix. Similarly, Hickery et al. [63] showed that IL-1 synthesis causes inhibition of proteoglycan sulfation in human articular chondrocytes and reported that IL-1-induced NO inhibits sulfation and alters the sulfation pattern of newly synthesized glycosaminoglycan chains [54]. Increased glycolysis induces high levels of lactate excreted to the ECM, decreased pH, and increased levels of metalloproteases such as MMP9 and ADAM10 [64]. However, ROS also contributes to cartilage degradation by mediating the activation of latent collagenase and upregulating the expression of genes coding for matrix metalloproteinases (MMP) [54,65], secreted by lysosomes and activated by Hpx or Hpx domains [64,66,67]. Moreover, an increase in the levels of cathepsins excreted by lysosomes [54] results in significant changes in various collagen types. Collagens I, II, and XI [42,68], as well as the conversion of collagen II to I [12,42,69], have been described in MPS IVA. In our study, we observed high levels of expression of proteoglycans, including KS (fibromodulin (Fmod) and Lumican (Lum)). In the context of increased ROS expression, the sorting receptor Sorl1 (detected only in UNT mice) induces more lysosomal endocytosis activity while the hydrolysis of sphingomyelin to ceramide increases. This effect may be reflected in our data by the low levels of Asah1 enzyme detected in UNT mice [27,70]. Proteins that participate in collagen endocytosis, including fibromodulin, integrins, and caveolin-1, all showed altered expression in UNT mice, in line with other findings (reviewed in Rainero, 2016) [71].

To validate our results, we used a new proteomic quantitative measure to estimate the difference in protein levels between UNT and WT mice. This validation was performed using DDA data that were transformed into a spectral count using Scaffold software v 5.3.2. Next, proteins for which differential expression was observed were analyzed using DDA and SWATH-MS, resulting in the validation of 20% of proteins. Thus, 38 proteins were considered differentiated expressed in UNT versus WT mice. These DEPs were related to ATP/metabolic process, glycolysis and gluconeogenesis, the TCA cycle, and LDH expression in the mitochondrial membrane. Interestingly, previous studies have reported alterations in some proteins (Ldha and Ldhab) [18,19] that we found to be differentially expressed (upregulated in UNT) and validated by both techniques, suggesting that elevation of LDH may constitute an early biomarker of MPS IVA.

These potential biomarkers are related to KS accumulation, as their upregulation is linked to an increase in ROS levels [30]. Our findings identify several proteins that may constitute an excellent candidate for diagnostic, prognosis, and disease monitoring biomarkers, including MMP-9, collagen type II, and LDH.

LDH has significant potential as a biomarker as it interacted extensively with many of the proteins found to be dysregulated in our study (Appendix A). Ldha and Ldhb in particular exhibited many interactions with the proteins listed in Appendix A, including those related to TCA, OXPHOS, ROS, and even the cytoskeleton.

It should be noted that our analysis was performed using only four bone samples per group. Further studies with a larger sample size using a different technology (i.e., other omics technologies) will be necessary to corroborate the present findings and better understand the pathogenic mechanisms that occur in the bones of children with MPS IVA.

The literature contains abundant information about the iNOS pathway, which generates large amounts of NO, and studies in chondrocytes indicate that this pathway is altered in MPS IVA, which activates IL-1 and acts on the extracellular matrix. In the present study, we could not quantify alterations in IL-1 or IL-6 due to the proteomic technique used. Further studies using more significant numbers of samples will be required to identify the proteins affected in this pathology.

In summary, MPS IVA is caused by an accumulation of glycosaminoglycans, which leads to a significant deficit in carbohydrate recirculation (yellow route). This disruption affects all the cells’ energy production routes, generating an increase in extracellular lactate dehydrogenase. The search for alternative nutrients causes protein dysregulation across these pathways, inducing high oxidative stress (ROS), which will also be generated in different cellular locations, such as the mitochondria and lysosomes. The inhibition of autophagy and certain proteasome proteins will alter the main lysosomal pathway. To counter these effects, TFEB activates compensatory mechanisms, which activate additional pathways, such as STAT3, which in turn activates interleukins and cytokines. There will also be an increase in activity directing proteins and vesicles towards the lysosome (purple route). Alterations to calcium activity will interfere with cellular functions, such as mobility and stability (cytoskeleton), hindering the transport of calcium phosphate needed for bone tissue remodeling (blue route). Finally, the decreased expression of certain proteins could lead to typical manifestations of Morquio A syndrome, such as skeletal deformities, delayed ossification in the growth plates of long bones, and reduced bone mineral density (Figure 14).

## 4. Material and Methods

### 4.1. Experimental Design and Statistical Analysis

Analyses were performed using samples from 4 untreated MPS IVA (UNT) and 4 WT male mice. The mice were provided by Dr. Tomatsu of Nemours Children’s Health (Wilmington, DE, USA), with prior authorization from that institution’s animal ethics committee (RSP19-12482-002). Mice were not subjected to any intervention and were euthanized at 6 weeks of age. The mice used carried the specific GALNS gene disruption in exon 2 [17].

All experiments were conducted using the TripleTOF 6600+ system (Sciex, Framingham, MA, USA). All mass spectrometry proteomics data have been deposited to the ProteomeXchange Consortium via the PRIDE partner repository with the data set identifier PXD042166. Additional information on the samples analyzed, data acquisition, and processing procedures can be found below.

SWATH analysis was performed using MarkerView 1.3.1; GraphPad Prism 6.01 software (GraphPad Software v 9.0.0, San Diego, CA, USA) was used to generate SWATH box plots, and spectral counting analysis was performed using Scaffold™ 5.3. In the SWATH analysis, the summed averaged areas of all transitions for each protein in each sample were analyzed using Student’s *t*-test to determine the extent to which each variable could distinguish between the two groups, and the result was reported as a *p*-value. Fisher’s exact test and the Benjamini–Hochberg multiple correction test were used for quantification and fold-change (FC) determination for spectral counting comparison between groups. In both cases, a protein was considered differentially expressed when *p* was < 0.05 and the FC > 1.5 or <0.6, although only proteins with an FC > 3 are shown in the figures.

In SWATH analysis, MLR (most likely ratio) normalization was performed after statistical analysis to control for possible uneven sample loss across the different samples during the sample preparation process [72,73]. The mean mass spectrometry peak area for each protein was determined for each sample, and Student’s *t*-test (MarkerView software v 1.3.1, Sciex, Framingham, MA, USA) was performed to compare between samples based on the averaged total area of all transitions for each protein. The *t*-test result (*p*-value) indicates how well each variable distinguishes between the two groups. Candidate proteins were selected based on *t*-test results (*p* < 0.05 and FC [increase or decrease] >1.5 or <0.6).

### 4.2. Animal Model and Bone Protein Extraction

Analyses were performed using 4 untreated MPS IVA (UNT) and 4 WT male mice. Mice were provided by Dr. Tomatsu of Nemours Children’s Health (Wilmington, DE, USA), with prior authorization from The Institutional Animal Care and Use Committee (IACUC), approval by Paul T. Fawcett, Ph.D. (RSP19-12482-002). Mice were not subjected to any intervention and were euthanized at 6 weeks of age. Femurs with intact femoral caps were collected, and samples were placed on dry ice. To prepare samples, a radioimmunoprecipitation assay (RIPA) buffer was added, and the resulting mixture was placed in tubes containing 2 mm porcelain beads and homogenized for 30 s using a PowerLizerco. After centrifugation for 30 min at 4 °C, the pellet and beads were separated. Each sample was analyzed for total BCA protein and N-acetylgalactosamine-6-sulfate sulfatase enzymatic activity. The remaining samples were sent for analysis at the proteomic analysis platform of the investigation department (IDIS) of the University Clinical Hospital of Santiago de Compostela.

Samples were analyzed using 2 distinct but complementary technologies: DDA (data-dependent analysis) and SWATH-MS (IDA, information-dependent acquisition) (Appendix A). One of the advantages of utilizing a label-free LC-MSMS workflow such as SWATH-MS is the sensitivity of protein detection and quantification offered. Traditional LC-MSMS workflows based on DDA increase the overall identification of proteins but suffer from a reduction in the number of identifiable peptides across multiple runs [74,75]. Therefore, SWATH-MS constitutes an attractive solution to improve label-free quantification and has been used for biomarker identification in other rare diseases [18,19].

To understand the interactions between proteins with altered expression in bone samples, we analyzed all SWATH-MS and DDA data, grouping proteins according to known functions, as follows: (i) energy metabolic pathways (e.g., glycolysis, tricarboxylic acid cycle, β-oxidation, and OXPHOS); (ii) mitochondrial function (i.e., oxidative stress, calcium-regulated mitochondrial electrical potential channels, apoptosis, compensatory mechanisms of oxidative stress, and DNA damage prevention); (iii) protein production (ribosomes), cytoskeleton, proteosomes, vesicular trafficking, and lysosomes; (iv) involvement in ECM function; and (v) other proteins of interest.

**Proteomic Analysis by TripleTOF 6600 LC-MSMS:** (see Appendix A for further information).

**Protein Digestion:** For protein identification, equal amounts of protein from each femur sample (n = 4 per group) were loaded onto a 10% SDS-PAGE gel. The resulting condensed protein bands [76,77] underwent gel digestion using Trypsin [78] and were processed as described in the Appendix A and previously by our group [79].

**LC-MSMS in Data-Dependent Acquisition Mode, Shotgun Analysis, and Spectral Count Transformation:** Digested peptides from each mouse femur sample were separated using reverse phase chromatography as described previously [79] and in the Appendix A. Semi-quantification by spectral counting was performed using Scaffold software (version 5.0.1; Proteome Software Inc., Portland, OR, USA), as described previously [80]. MSMS-based peptide and protein identifications performed in ProteinPilot 5.0.1 software [81] were transformed in .mzid format and validated using Scaffold software. Peptide identifications were accepted if they could be established at >95.0% probability by the Percolator posterior error probability calculation. Protein identifications with a probability < 99.0% and at least two identified peptides were accepted. Protein probabilities were assigned by the Protein Prophet algorithm. Proteins that contained similar peptides and could not be differentiated based on MSMS analysis alone were grouped to satisfy the principles of parsimony [82,83,84].

**Protein Quantification by SWATH-MS Analysis [85,86,87,88]:** To build the MSMS spectral libraries, peptide solutions were analyzed by shotgun data-dependent acquisition (DDA) using micro-LC–MSMS, as described in the Appendix A and previously by our group [18,76]. The MSMS spectra of the identified peptides were then used to generate the spectral library for SWATH peak extraction using the add-in for PeakView Software (version 2.2, Sciex), MSMS^ALL^ with SWATH Acquisition MicroApp (version 2.0, Sciex). Peptides with a confidence score >99% (obtained from the ProteinPilot database search) were included in the spectral library. For relative quantification by SWATH-MS analysis, SWATH-MS acquisition was performed on a TripleTOF 6600 LC–MS–MS system (Sciex) using SWATH mode. The acquisition mode consisted of a 250 ms survey MS scan from 400 to 1250 *m*/*z*, followed by an MSMS scan from 100 to 1500 *m*/*z* (25 ms acquisition time) of the top 65 precursor ions from the survey scan, for a total cycle time of 2.8 s. The fragmented precursors were then added to a dynamic exclusion list for 15 s. Any singly charged ions were excluded from the MSMS analysis. Targeted data extraction from the SWATH MS runs was performed by PeakView v.2.2 (Sciex, Redwood city, CA, USA) using the SWATH-MS Acquisition MicroApp v.2.0 (Sciex, USA). Data were processed using the spectral library created from DDA. SWATH-MS quantization was attempted for all proteins in the ion library that were identified by ProteinPilot^TM^ 5.0.1 with a false discovery rate (FDR) <1%. PeakView computed an FDR and a score for each assigned peptide based on the chromatographic and spectra components: only peptides with an FDR < 1%, 10 peptides, and 7 transitions per peptide were used for protein quantization. The integrated peak areas were processed by MarkerView software version 1.3.1 (Sciex, USA) for a data-independent method for relative quantitative analysis. A most likely ratio normalization was performed to control for possible uneven sample loss across the different samples during the sample preparation process. Unsupervised multivariate statistical analysis using PCA was performed to compare data across samples.

### 4.3. Functional and Pathway Analysis

Pathway analysis was performed using Reactome (https://reactome.org/ accessed on 20 July 2023), which applies a statistical (hypergeometric distribution) test to determine whether specific pathways are over-represented (enriched) and produces a probability score, which is corrected for FDR using the Benjamini–Hochberg method. The most enriched pathways were represented using Reactome pathway diagrams. Protein interactions were evaluated using STRING (https://string-db.org/ accessed on 25 July 2023)), applying a minimum required interaction score of PPI = 0.9 (protein-protein interaction) and an FDR < 0.05. Venn diagrams were generated using http://www.interactivenn.net/ accessed on 2 June 2023) and box plots using GraphPad Prism 9. Statistical analyses were performed using Markerview v 1.3.1 or Scaffold software v 5.2.2. Volcano plots and box plots were generated using GraphPad Prism 9.0.0.

### 4.4. In Silico Analysis

Proteins for “in silico” analysis were selected using two distinct approaches.

The first approach was based on SWATH-MS analyses of leukocytes and fibroblasts previously published by our group [18,19]. The second approach involved validating proteomic data sets taken from published analyses of human bone [20,21]. Few such analyses have been published; most correspond to bone marrow samples. For our validation, we selected 2 data sets, corresponding to bone and bone marrow sample data sets. The data set that was extracted for our analysis fulfilled the following 3 quality assessment criteria: (1) identification and description of the extraction method used; (2) description of the sampling process used; (3) use of a proteomic technology similar to SWATH-MS.

### 4.5. Translational Significance

In our study, we identify and quantify (Shotgun-DDA and SWATH-IDA) the proteins present in WT and UNT mouse bones. We found an accumulation of ROS due to excessive glycolysis, LDH upregulation, and accumulation of substrates in the lysosome: an increase in ROS levels triggers the activation of pathways such as STAT3 signaling, which compensates for deficient autophagy and activates cytokines and interleukins. We propose LDH as a valuable biomarker for diagnosing and monitoring disease progression. The translational significance of this research is not only in identifying novel protein biomarkers of MPS IVA using proteomic technologies but also in providing more information about the pathology mechanism.

## 5. Conclusions

The pathological features of MPS IVA resemble those of other skeletal dysplasias and bone diseases that occur at later ages. However, MPS IVA can also involve ROS accumulation due to excessive glycolysis, upregulation of LDH, and accumulation of substrates in the lysosome. This increase in ROS levels triggers the activation of pathways such as STAT3 signaling, which compensates for deficient autophagy and activates cytokines and interleukins. In MPS IVA, this process becomes chronicled due to the accumulation of deposits that are not eliminated in the lysosomes.

In MPS IVA, dysfunction at the level of the mitochondria and cytoskeleton can be due to calcium deficiency, possibly due to high bone resorption, poor bone mineralization, and excessive ROS activation, which facilitates active-matrix degradation. These alterations can lead to a matrix composition change, mainly due to defective bone regeneration.

Our proteomic technique analyses identify MMP-9 and collagen type II as candidate biomarkers of MPS IVA, suggesting that LDH may constitute a new candidate biomarker. LDH is upregulated in various human cell types (see our studies) [18,19] and downregulated in response to enzyme replacement therapy. All these candidate biomarkers detected in mouse bone samples in the present study have also been found to be altered using proteomic techniques in human bone and bone marrow samples. This biomarker may constitute a useful biomarker for diagnosis and monitoring disease progression of therapeutic efficacy in MPS IVA disease.

## Figures and Tables

**Figure 1 ijms-25-03232-f001:**
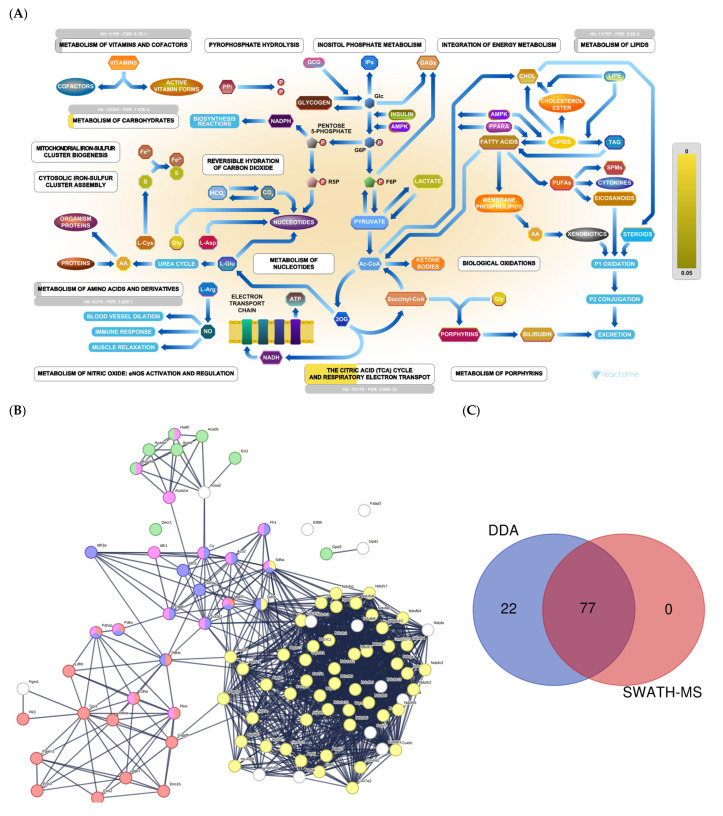
(**A**) Metabolic pathways: Reactome pathway enrichment analysis of DEPs. (**B**) String interaction analysis of DEPs. Colored dots represent proteins implicated in glycolysis and gluconeogenesis (red); fatty acid β-oxidation (green); the electron transport chain (yellow); TCA cycle (blue); amino acid metabolism (pink). (**C**) Venn diagram showing the overlap of DEPs identified using SWATH-MS and all proteins identified by DDA.

**Figure 2 ijms-25-03232-f002:**
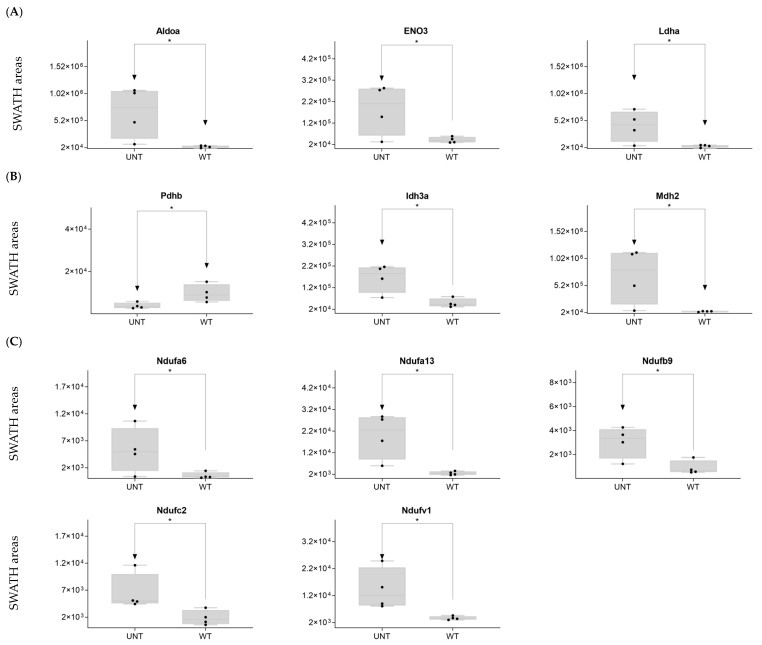
(**A**–**C**) Relative expression levels of DEPs (*p* < 0.05; FC > 3) implicated in glycolysis in UNT and WT mice. (**A**) TCA cycle. (**B**) Krebs cycle. (**C**) Black dots represent the SWATH-MS area values of each sample, per group. * *p* < 0.05.

**Figure 3 ijms-25-03232-f003:**
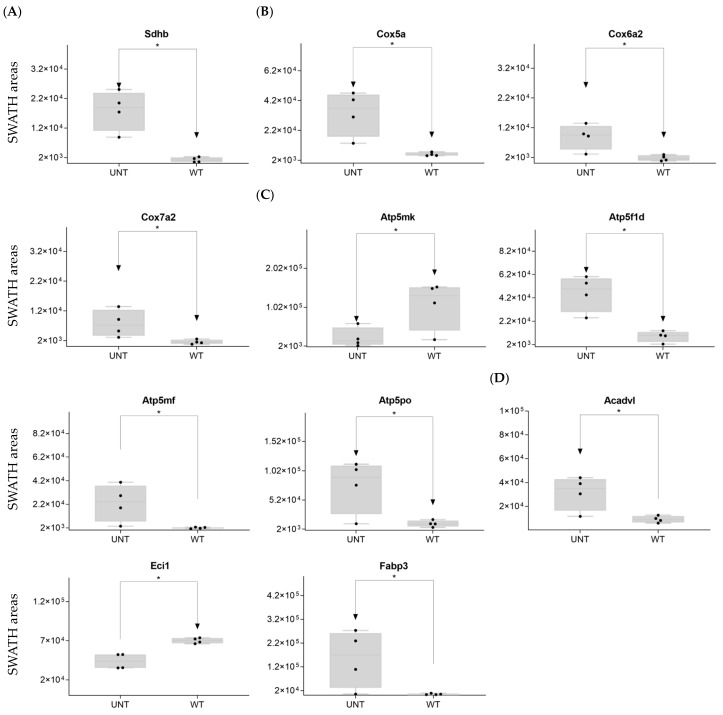
(**A**–**D**) Relative expression levels of DEPs (*p* < 0.05; FC > 3) implicated in OXPHOS in UNT and WT mice. (**A**) Mitochondria complex. (**B**) Mitochondria complex V, ATP production. (**C**) β-oxidation. (**D**) Black dots represent the SWATH-MS area values of each sample, per group. * *p* < 0.05.

**Figure 4 ijms-25-03232-f004:**
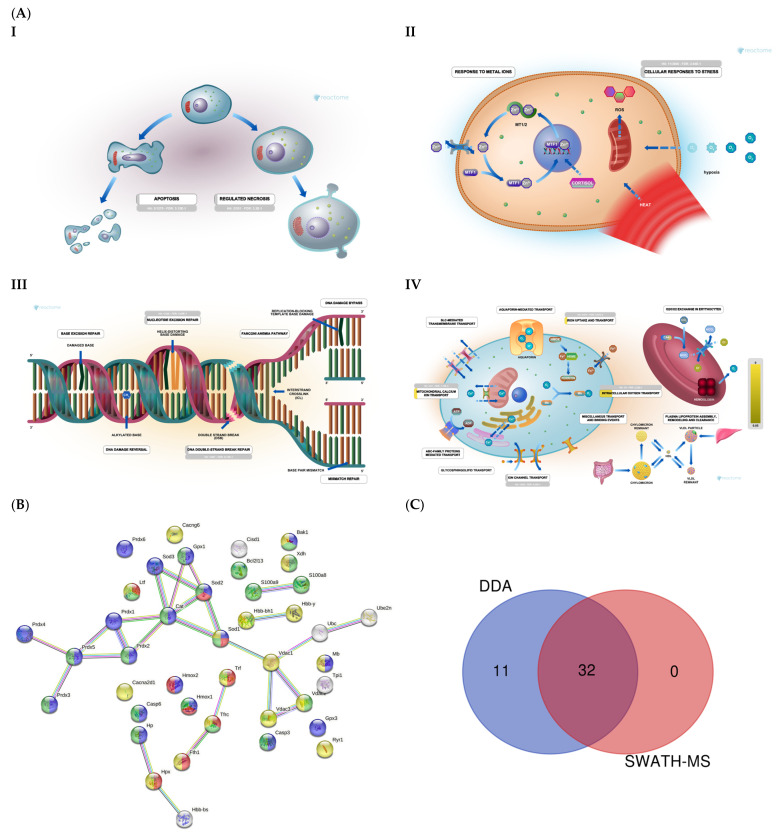
(**A**) Mitochondrial function: Reactome pathway enrichment analysis of DEPs related to apoptosis (I); cellular response to stress (II); DNA repair (III); iron transport (IV). (**B**) STRING interaction analysis of DEPs. Colored dots represent proteins related to iron transport (red); cell death (green); transport (yellow); oxidative stress (blue). (**C**) Venn diagram showing the overlap of DEPs identified using SWATH-MS and DDA.

**Figure 5 ijms-25-03232-f005:**
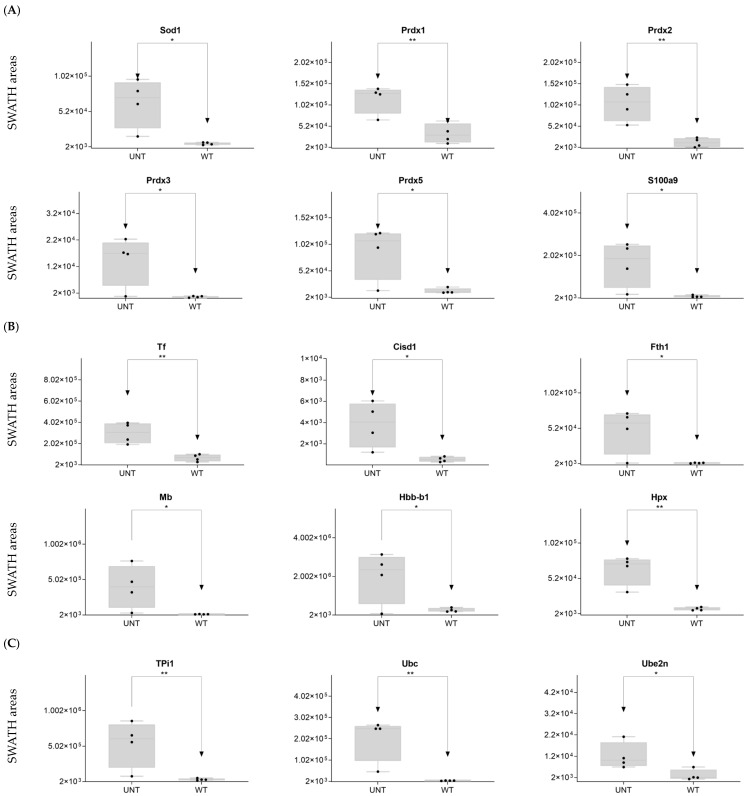
Relative expression levels in UNT and WT mice of DEPs (*p* < 0.05; FC > 3) related to oxidative stress (**A**); iron transport (**B**); DNA damage prevention (**C**). Black dots represent the SWATH-MS area values of each sample, per group. * *p* < 0.05; ** *p* < 0.01.

**Figure 6 ijms-25-03232-f006:**
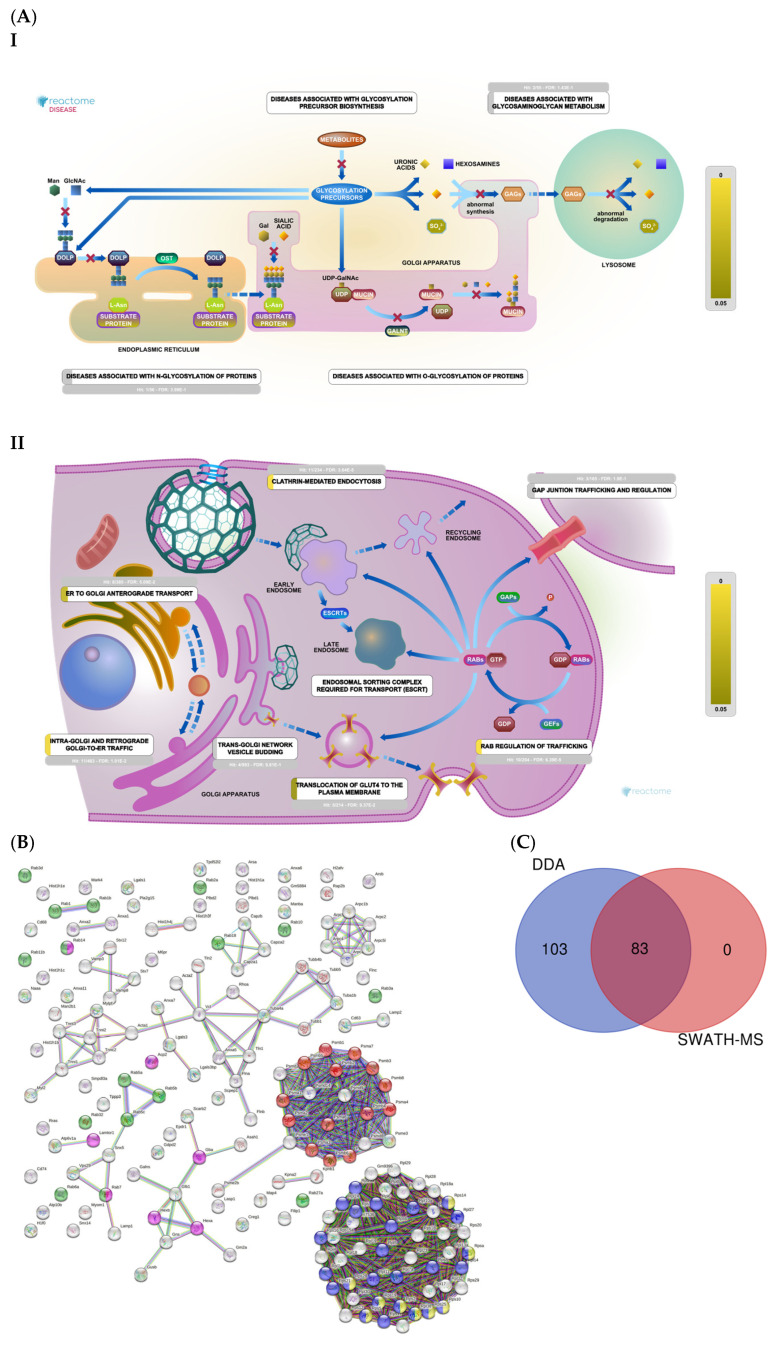
(**A**) Histones, ribosomes, proteasomes, vesicular transport, and lysosomes: Reactome pathway enrichment analysis of DEPs related to lysosomes (I); and vesicular transport (II). (**B**) Colored dots represent proteins related to proteasomes (red); rab protein signal translation (green); small ribosomal subunits (yellow); ribosomal biogenesis (blue); lysosomes (pink). (**C**) Venn diagram showing the overlap of DEPs identified using SWATH-MS and DDA.

**Figure 7 ijms-25-03232-f007:**
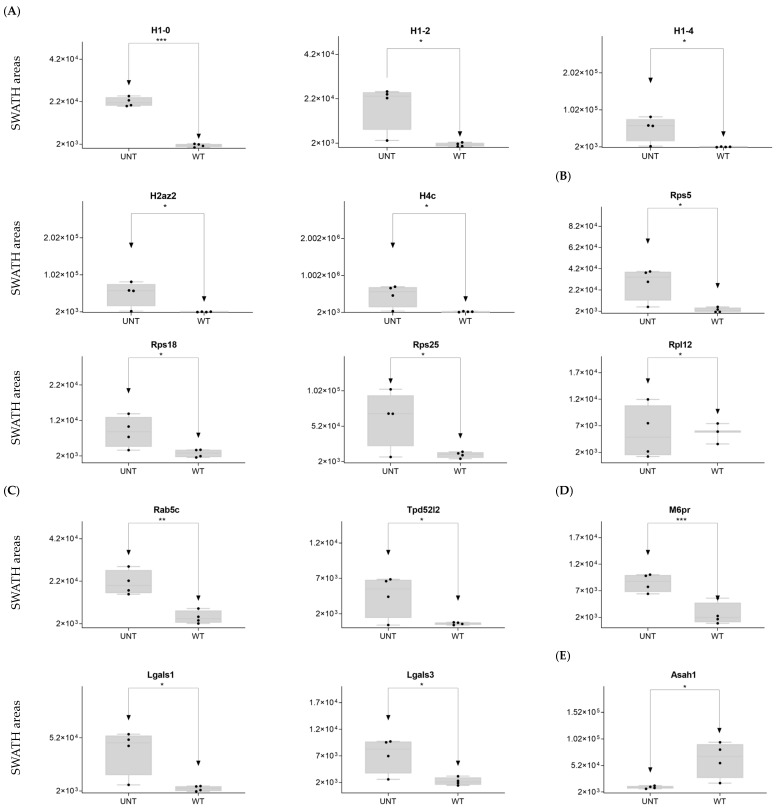
Relative expression levels in UNT and WT mice of DEPs (*p* < 0.05; FC > 3) related to histones (**A**); ribosomal proteins (**B**); vesicular transport (**C**); lysosomal membrane (**D**); whole lysosome (**E**). Black dots represent the SWATH-MS area values of each sample, per group. * *p* < 0.05; ** *p* < 0.01; *** *p* < 0.001.

**Figure 8 ijms-25-03232-f008:**
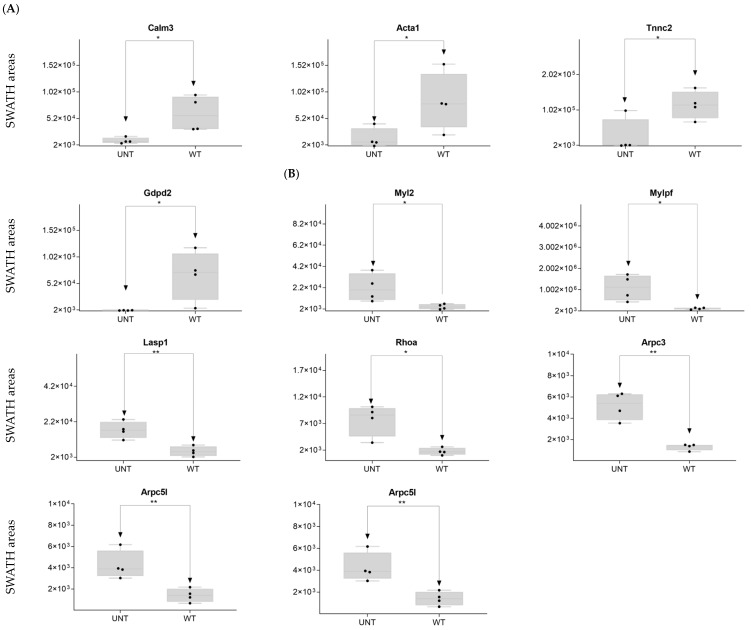
Relative expression levels in UNT and WT mice of DEPs (*p* < 0.05; FC > 3) related to the cytoskeleton. Cytoskeletal proteins upregulated in WT (**A**) and UNT (**B**) mice. Black dots represent the SWATH-MS area values of each sample, per group. * *p* < 0.05; ** *p* < 0.01.

**Figure 9 ijms-25-03232-f009:**
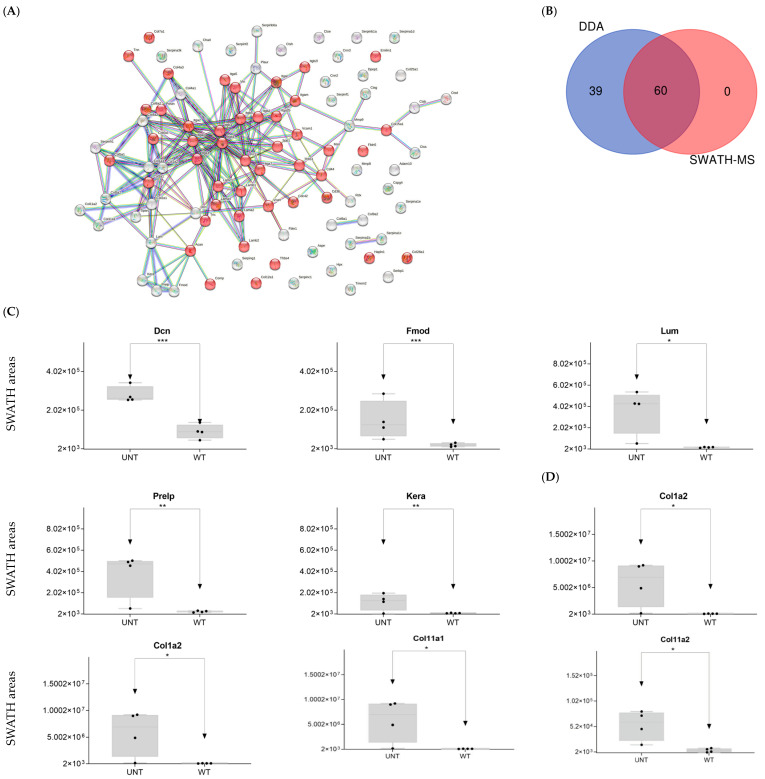
(**A**) STRING interaction analysis of DEPs. Red dots indicate proteins related to cell adhesion. (**B**) Venn diagram showing overlap of DEPs across the two proteomic techniques used. (**C**) Relative expression levels in UNT and WT mice of DEPs for proteoglycans. (**D**) Relative expression levels in UNT and WT mice of DEPs for collagens (*p* < 0.05; FC > 3) related to proteoglycans. * *p* < 0.05; ** *p* < 0.01; *** *p* < 0.001.

**Figure 10 ijms-25-03232-f010:**
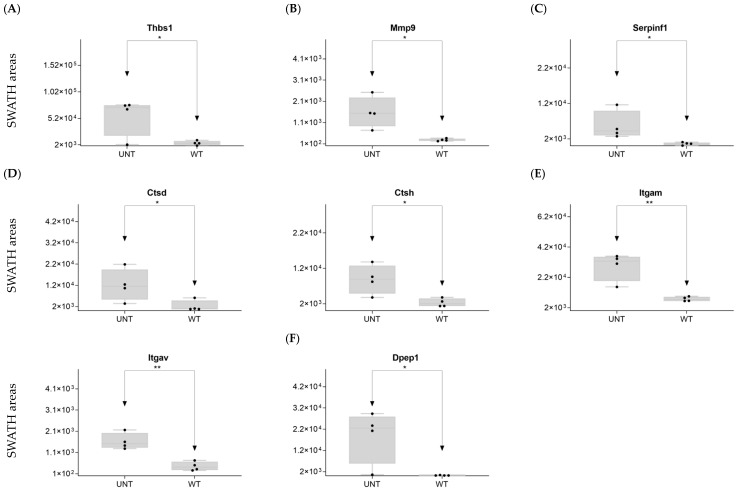
Relative expression levels in UNT and WT mice of DEPs (*p* < 0.05; FC > 3) related to fibrillar collagen (**A**); extracellular receptor matrix (**B**); extracellular matrix proteases (**C**); cathepsins; (**D**) integrins; (**E**); discoidin domain receptor (**F**). * *p* < 0.05; ** *p* < 0.01.

**Figure 11 ijms-25-03232-f011:**
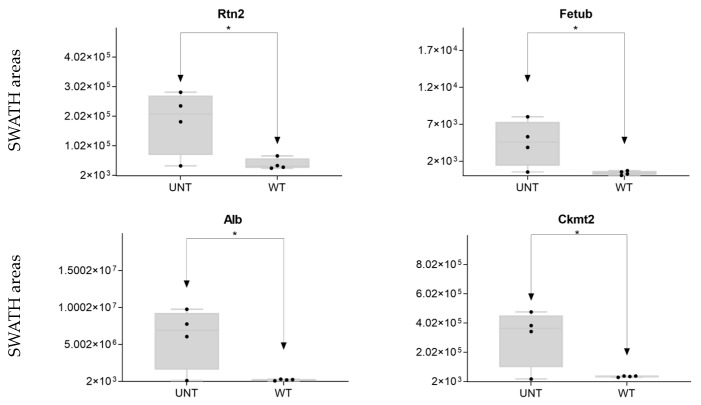
Other proteins of interest: relative expression levels of DEPs (* *p* < 0.05; FC > 3) in UNT and WT mice. Black dots represent the SWATH-MS area values of each sample, per group.

**Figure 12 ijms-25-03232-f012:**
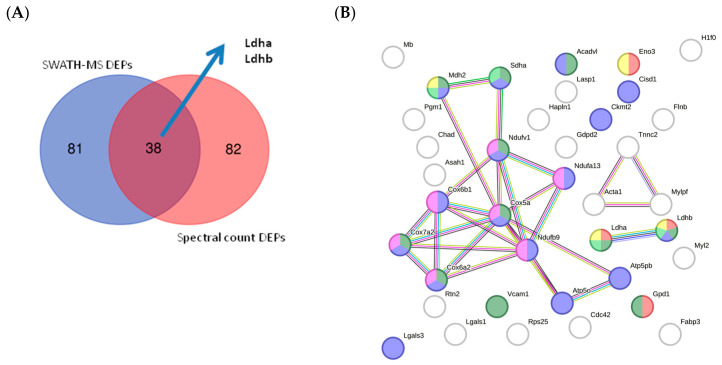
(**A**) Venn diagram showing the overlap of validated DEPs identified using each proteomic technique. (**B**) STRING interaction analysis of validated DEPs showing proteins related to ATP metabolic process (red); glycolysis and gluconeogenesis (yellow); the TCA cycle (light green); the mitochondrial membrane (blue); OXPHOS (pink); and ROS (dark green).

**Figure 13 ijms-25-03232-f013:**
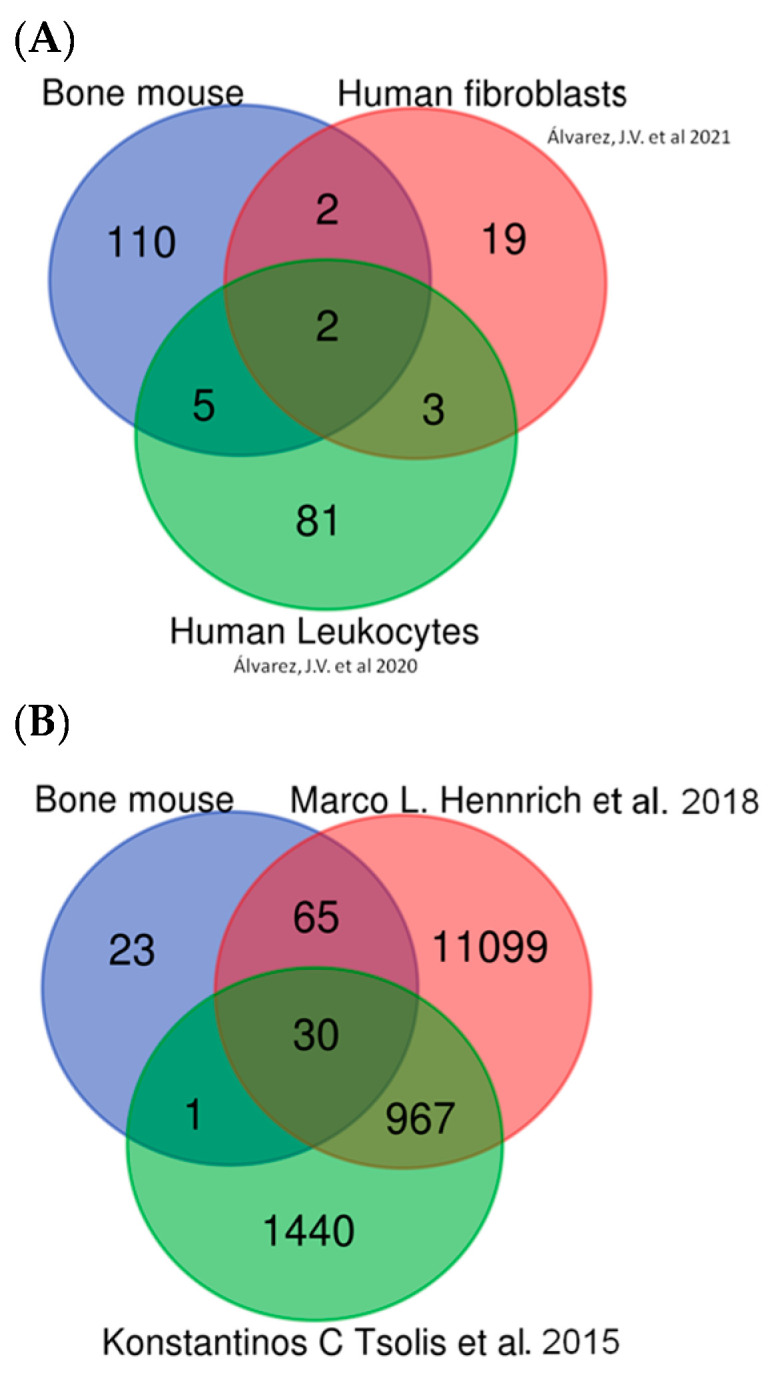
In silico validation. (**A**) Comparison between mouse DEPs and our previous papers in human samples [18,19]. (**B**) Comparison between mouse DEPs and other data sets in human bone and human bone marrow samples [20,21].

**Figure 14 ijms-25-03232-f014:**
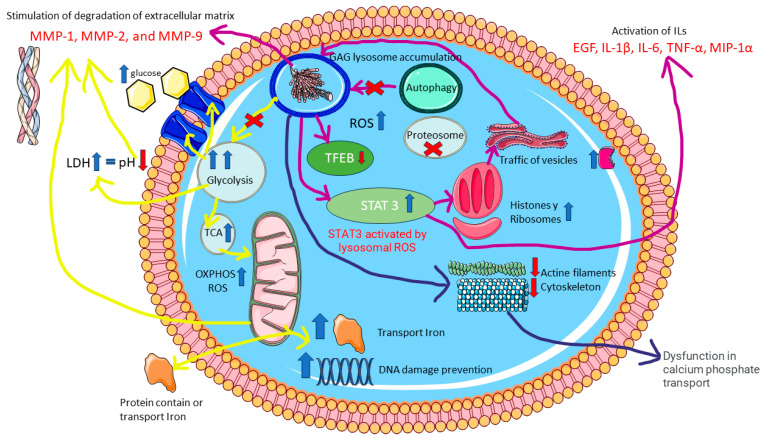
Proteomic dysregulation in MPS IVA. The figure highlights the main proteins, pathways, and cellular components involved.

## Data Availability

The mass spectrometry proteomics data have been deposited to the ProteomeXchange Consortium via the PRIDE partner repository with the data set identifier PXD042166.

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
