# Peer review of "Morquio A Syndrome: Identification of Differential Patterns of Molecular Pathway Interactions in Bone Lesions"

_ijms, 2024, doi:10.3390/ijms25063232_

Round 1

Reviewer 1 Report

Comments and Suggestions for Authors

alvarez et al studied molecular pathways in bone involvement in MPS IVA, using metabolimic assays. The study is very interesting and important for future therapeutic developments. It is well structured and presents and the results are very important. I have only few suggestions: 

- in the introduction the sentence "Although the clinical presentation... elucidated" should be moved before the previous paragraph

- discussion is very long. The second paragraph, on metabolomics techniques, could be simplified

Author Response

Review 1

Alvarez et al studied molecular pathways in bone involvement in MPS IVA, using metabolimic assays. The study is very interesting and important for future therapeutic developments. It is well structured and presents and the results are very important. I have only few suggestions: 

RESPONSE: We thank the Reviewer#1 for the positive view of our manuscript.

- in the introduction the sentence "Although the clinical presentation... elucidated" should be moved before the previous paragraph

RESPONSE: Thank the reviewer for this appreciation. We move this sentence before previous paragraph as you suggest.

- Ddiscussion is very long. The second paragraph, on metabolomics techniques, could be simplified

RESPONSE: Thank the reviewer for this appreciation. We simplified the second paragraph in the discussion section according your suggestions.

Reviewer 2 Report

Comments and Suggestions for Authors

The article by José V. Álvarez and colleagues is a very detailed and informative summary of "Morquio A Syndrome: Identification of Differential Patterns of Molecular Pathway Interactions in Bone Lesions." The article clearly lays out the background, methodology, results and discussions, and conclusions of the study in a comprehensive and easy-to-understand way. The authors further highlight the translational significance of this research in identifying novel protein biomarkers of MPS IVA using Proteomic technologies and gaining more information about the pathology mechanism.

The primary question addressed by this research article is: What are the differential patterns of molecular pathway interactions in bone lesions of Morquio A Syndrome? By comparing bone proteomes of MPS IVA mice and wild-type mice, the authors aimed to elucidate the molecular mechanisms underlying bone involvement in this rare autosomal recessive lysosomal storage disorder (LSD).

This study employs a novel approach by combining Dependent Data Analysis (DDA) and Sequential Window Acquisition of All Theoretical Fragment Ion Mass Spectra (SWATH-MS) proteomic techniques to investigate bone lesions in MPS IVA mice. This allows for a more comprehensive analysis of protein expression and potential pathway interactions compared to previous studies. The results obtained from the study suggest

the mechanisms underlying bone involvement in Mucopolysaccharidosis type IVA (MPS IVA; Morquio A syndrome), a poorly understood aspect of the disease. Identifying potential biomarkers (MMP-9 and collagen type II) suggests that (Lactate dehydrogenase) LDH could significantly impact diagnosis and monitoring.

This study offers mechanisms and insights that identify protein changes to specific pathways and cellular processes relevant to MPS IVA bone pathology. The combination of DDA and SWATH-MS techniques further strengthens the identification and quantification of differentially expressed proteins. 

The references appear appropriate and up-to-date and cover relevant MPS IVA and bone biology studies.

The following comments need to be addressed by the authors: 

  1. Authors need to use either KO or UNT throughout the study. E.g. In the graphical abstract, KO and UNT are mentioned in different places. This might create additional confusion. Similarly, this goes for the use of WT or Wt. 

  2. The authors should be consistent with the abbreviation of MSMS or MS2 (MS/MS) or MS2 (MSMS). 

  3. The language is clear and professional throughout, but a few minor grammatical errors in the main article and supplementary files could be smoothed out.

  4. The text must be clearly visible in Figures 1A and 1B, and authors must provide high-resolution images. 

  5. The font size on the X and Y axes needs to be readable in Figures 2, 3, 5, 7, 8, 10, and 11. The author needs to increase the font size. 

  6. In Figure 4A, the text is not visible at all. Authors need to mention the source of the figures. This also applies to Figure 6A. 

  7. In Figure 13B, instead of writing the protein's name in the figure, the author should provide a separate list of the protein in Table format. 

  8. The authors should discuss  “Proteome maps summary of deregulation proteins in MPS IVA” in the discussion section. There is no figure number or figure description for this. I recommend the authors either include a relevant figure number with a detailed description in the figure legend or provide a clear explanation within the text describing the figure's content and key findings. This will enhance the reader's understanding of the study's crucial results.

Author Response

Reviewer 2

The article by José V. Álvarez and colleagues is a very detailed and informative summary of "Morquio A Syndrome: Identification of Differential Patterns of Molecular Pathway Interactions in Bone Lesions." The article clearly lays out the background, methodology, results and discussions, and conclusions of the study in a comprehensive and easy-to-understand way. The authors further highlight the translational significance of this research in identifying novel protein biomarkers of MPS IVA using Proteomic technologies and gaining more information about the pathology mechanism.

 The primary question addressed by this research article is: What are the differential patterns of molecular pathway interactions in bone lesions of Morquio A Syndrome? By comparing bone proteomes of MPS IVA mice and wild-type mice, the authors aimed to elucidate the molecular mechanisms underlying bone involvement in this rare autosomal recessive lysosomal storage disorder (LSD). 

This study employs a novel approach by combining Dependent Data Analysis (DDA) and Sequential Window Acquisition of All Theoretical Fragment Ion Mass Spectra (SWATH-MS) proteomic techniques to investigate bone lesions in MPS IVA mice. This allows for a more comprehensive analysis of protein expression and potential pathway interactions compared to previous studies. The results obtained from the study suggest

the mechanisms underlying bone involvement in Mucopolysaccharidosis type IVA (MPS IVA; Morquio A syndrome), a poorly understood aspect of the disease. Identifying potential biomarkers (MMP-9 and collagen type II) suggests that (Lactate dehydrogenase) LDH could significantly impact diagnosis and monitoring.

 This study offers mechanisms and insights that identify protein changes to specific pathways and cellular processes relevant to MPS IVA bone pathology. The combination of DDA and SWATH-MS techniques further strengthens the identification and quantification of differentially expressed proteins. 

The references appear appropriate and up-to-date and cover relevant MPS IVA and bone biology studies.

 RESPONSE: We thank the Reviewer#2 for the positive view of our manuscript. We also thank for his/her comments.

The following comments need to be addressed by the authors: 

  1. Authors need to use either KO or UNT throughout the study.  E.g. In the graphical abstract, KO and UNT are mentioned in different places. This might create additional confusion. Similarly, this goes for the use of WT or Wt. 

RESPONSE: Thank the reviewer for this appreciation. We have changed this, thank you. We now named UNT and WT to the groups. Therefore, picture of graphical abstract has been replaced with the correct name too

  1. The authors should be consistent with the abbreviation of MSMS or MS2 (MS/MS) or MS2 (MSMS). 

RESPONSE: Thank the reviewer for this appreciation. We have changed this, thank you. We now named MSMS

  1. The language is clear and professional throughout, but a few minor grammatical errors in the main article and supplementary files could be smoothed out.

RESPONSE: Thank the reviewer for this appreciation.  We have checked grammatical errors in the main article and supplementary files and this article was revised by a native speaker.

  1. The text must be clearly visible in Figures 1A and 1B, and authors must provide high-resolution images. 

RESPONSE: We improve the quality of the both figures downloading the figure in format PNG from Reactome page (https://reactome.org/?ref=blog.opentargets.org) and in hire quality the figure from the string. Moreover this last figure was bigger than previously image to contribute to a better letter understand

  1. The font size on the X and Y axes needs to be readable in Figures 2, 3, 5, 7, 8, 10, and 11. The author needs to increase the font size. 

RESPONSE: We increase both letter sizes in all figures

  1. In Figure 4A, the text is not visible at all. Authors need to mention the source of the figures. This also applies to Figure 6A. 

RESPONSE: Thank the reviewer for his/her appreciations. We improve the quality of the both figures downloading the figure in format PNG from Reactome page (https://reactome.org/?ref=blog.opentargets.org)

  1. In Figure 13B, instead of writing the protein's name in the figure, the author should provide a separate list of the protein in Table format.

RESPONSE: Thank the reviewer for his/her appreciations. We include all the proteins that are found in all comparison as supplementary table S7. We indicate this modification in the text.

  1. The authors should discuss  “Proteome maps summary of deregulation proteins in MPS IVA” in the discussion section. There is no figure number or figure description for this. I recommend the authors either include a relevant figure number with a detailed description in the figure legend or provide a clear explanation within the text describing the figure's content and key findings. This will enhance the reader's understanding of the study's crucial results.

RESPONSE. We include this paragraph in the test. In summary MPS IVA is caused by an accumulation of glycosaminoglycans, which leads to a significant deficit in carbohydrate recirculation (yellow route). This disruption affects all the cell’s energy production routes, generating an increase in extracellular lactate dehydrogenase. The search for alternative nutrients causes protein dysregulation across these pathways, inducing high oxidative stress (ROS), which will also be generated in different cellular locations, such as the mitochondria and lysosomes. The inhibition of autophagy and certain proteasome proteins will alter the main lysosomal pathway. To counter these effects, TFEB activates compensatory mechanisms, which activate additional pathways, such as STAT3, which in turn activates interleukins and cytokines. There will also be an increase in activity directing proteins and vesicles towards the lysosome (purple route). Alterations to calcium activity will interfere with cellular functions, such as mobility and stability (cytoskeleton), hindering the transport of calcium phosphate needed for bone tissue remodelling (blue route). Finally, the decreased expression of certain proteins could lead to typical manifestations of Morquio A syndrome, such as skeletal deformities, delayed ossification in the growth plates of long bones, and reduced bone mineral density (Figure 14).

Reviewer 3 Report

Comments and Suggestions for Authors Good paper. There are many experiments to support the conclusions. The only request is to explain in more detail the role of LDH and its association with the study of different proteins. It may be useful to specify for each protein evaluated its binding to LDH. Make a summary table of LDH and its correlation with the different proteins

Author Response

Good paper. There are many experiments to support the conclusions. The only request is to explain in more detail the role of LDH and its association with the study of different proteins. It may be useful to specify for each protein evaluated its binding to LDH. Make a summary table of LDH and its correlation with the different proteins

RESPONSE: Thank the reviewer for his/her appreciations. We include a paragraph in the discussion and a new supplementary table S8 with the proteins that interact with LDHa or LDHb and was found deregulated in our SWATH analysis.

LDH has significant potential as a biomarker as it interacted extensively with many of the proteins found to be dysregulated in our study (Supplementary Table S8). Ldha and Ldhb in particular exhibited many interactions with the proteins listed in Table S8, including those related to TCA, OXPHOS, ROS, and even the cytoskeleton.

Round 2

Reviewer 2 Report

Comments and Suggestions for Authors

The authors have addressed my comments.